# A Modified-Herringbone Micromixer for Assessing Zebrafish Sperm (MAGS)

**DOI:** 10.3390/mi14071310

**Published:** 2023-06-26

**Authors:** Jorge A. Belgodere, Mustafa Alam, Valentino E. Browning, Jason Eades, Jack North, Julie A. Armand, Yue Liu, Terrence R. Tiersch, W. Todd Monroe

**Affiliations:** 1Department of Biological and Agricultural Engineering, Louisiana State University and Agricultural Center, Baton Rouge, LA 70803, USA; jbelgo1@lsu.edu (J.A.B.); malam19@lsu.edu (M.A.); vdudz1234@gmail.com (V.E.B.); jason.eades.10@gmail.com (J.E.); jnorth9@lsu.edu (J.N.); jarman4@lsu.edu (J.A.A.); yliu@agcenter.lsu.edu (Y.L.); 2Aquatic Germplasm and Genetic Resources Center, School of Renewable Natural Resources, Louisiana State University Agricultural Center, Baton Rouge, LA 70820, USA; ttiersch@agcenter.lsu.edu

**Keywords:** micromixer, high-dilution, microfabricated, PDMS, zebrafish

## Abstract

Sperm motility analysis of aquatic model species is important yet challenging due to the small sample volume, the necessity to activate with water, and the short duration of motility. To achieve standardization of sperm activation, microfluidic mixers have shown improved reproducibility over activation by hand, but challenges remain in optimizing and simplifying the use of these microdevices for greater adoption. The device described herein incorporates a novel micromixer geometry that aligns two sperm inlet streams with modified herringbone structures that split and recombine the sample at a 1:6 dilution with water to achieve rapid and consistent initiation of motility. The polydimethylsiloxane (PDMS) chip can be operated in a positive or negative pressure configuration, allowing a simple micropipettor to draw samples into the chip and rapidly stop the flow. The device was optimized to not only activate zebrafish sperm but also enables practical use with standard computer-assisted sperm analysis (CASA) systems. The micromixer geometry could be modified for other aquatic species with differing cell sizes and adopted for an open hardware approach using 3D resin printing where users could revise, fabricate, and share designs to improve standardization and reproducibility across laboratories and repositories.

## 1. Introduction

The increased utilization of aquatic species as biomedical research models has highlighted the importance of evaluating sperm from these species for research applications as well as the development of germplasm repositories. Sperm motility assessment is the leading indicator of quality [1] but it poses a challenge due to the short duration (5–20 s) of peak sperm motility observed in many freshwater species [2]. The consistency of motility analysis following sperm activation has been improved with standardized chambers [3] and automated imaging via computer-assisted sperm analysis (CASA) [4]. However, the variation in user techniques of activation and a lack of standardization [5] yield a low reproducibility of research [6]. 

Microfluidics, the manipulation of small fluid volumes, has had a profound impact on many aspects of sperm analysis and selection [7]. The small volumes (µL) associated with sperm samples are readily assayed on-chip, with advantages such as low sample consumption, low cost, high portability, and high standardization. Microdevices have been widely used in human fertility research and sperm-related disease diagnosis, with commercial products approved by the US Food and Drug Administration (FDA) [8,9,10]. Most applications for microfluidic devices are focused on the isolation of sperm from cellular debris [11,12,13] or the selection of motile sperm for subsequent in vitro fertilization [14,15]. However, the potential of microfluidic chips to assist sperm quality analysis has not been well recognized in the field of aquaculture and fisheries. 

Given the aforementioned advantages and applications of microfluidics, the mixing of fluid streams in microdevices is a challenge, as fluid flows are laminar, such that mixing at these low Reynolds (Re) numbers is dependent on the speed of diffusion of the molecular species [16,17]. Microfluidic-based mixing geometries or micromixers have been extensively reviewed [16,18]. Micromixers can be classified as active, which requires external energy or passive, which utilizes only microchannel geometries to induce mixing. These methods can induce rapid mixing that enhances molecular diffusion and creates a fluid regime not limited by low flow rates or low Re values [19]. Active micromixers utilize energy sources that include electrical [20], pressure [16], magnetic [21], and acoustic [22] and can achieve high performance, flexible parameter selection, and high mixing efficiencies, but care should be taken to not disturb biological samples with increasing fluid temperatures [23,24]. Active mixers for sperm analysis have been developed, where artificial cilia were used to mix and activate zebrafish (*Danio rerio*) sperm [25,26].

Passive micromixers have no energy requirements but instead utilize unique channel features such as curved or serpentine channel shapes [27,28], flow splitting and recombining obstacles [29], and baffles or cavities that force fluid paths that increase mixing efficiency [30,31,32]. Herringbone micromixer geometries, named for their resemblance to fish bone patterns, have been widely cited and adapted for efficient mixing and low-pressure drop in the microchannel, as the cavities do not directly impede flow [33,34,35]. The first report of microfluidic mixing for zebrafish sperm activation utilized a two-inlet herringbone mixer that emptied into a separate and standardized viewing chamber for motility analysis [36]. Following that, a novel monolithic curved-channel mixer was developed that did not require the two-layer photolithography used to fabricate the herringbone structures [37]. This micromixer was incorporated into a device that accomplished micromixing with zebrafish sperm cell activation more rapidly and with less deviation than manual mixing by hand [38]. While each of these devices demonstrated on-chip activation, neither had a micromixer capable of achieving a larger than 1:3 dilution ratio, which has been reported as a critical factor in evaluating motility [39]. Most reported microdevices utilizing high-dilution or combinatorial mixing methods are focused on biochemical applications [40,41] rather than for use with live cells. In the microdevices where cells travel through the micromixers or diluting arrays, they have been designed for disruption such as cell lysis to access cytosolic contents [42] or cell removal from microcarrier beads [43].

The goal of this work was to design, simulate, fabricate, and evaluate a micromixer for assessing zebrafish sperm quality (MAGS) using passive micromixer geometries (Figure 1). The MAGS device contained a novel, high-dilution (1:6) inlet configuration that aligned incoming sperm sample streams with modified herringbone geometries in the main channel. The device can accommodate samples manually using a micropipette or can be automated using a syringe pump The specific objectives were to (1) design and simulate different micromixer and inlet configurations to compare mixing efficiencies against standard herringbone mixer geometries and traditional T-channel inlet designs, (2) utilize soft lithography to fabricate micromixer prototypes and validate simulations of mixing, and (3) evaluate the practical use of the micromixer in activating zebrafish sperm to allow interrogation of a single sample at multiple points along the channel length in a standard CASA system.

## 2. Materials and Methods

### 2.1. Device Design of Inlet and Passive Micromixers

The proposed micromixer incorporated a split inlet channel that aligned the sample streams with symmetric v-shaped herringbone features in the main channel (Figure 2). The proposed inlet configuration included 2 sperm inlets and 3 diluting water inlet streams sized to achieve a large dilution ratio for optimal zebrafish sperm activation [39]. The vertices of the v-shaped herringbone structures were positioned to ensure equal splitting of both sperm streams and were symmetrical across the channel for homogenous mixing. The initial flight of herringbones acted to split the fluid streams laterally towards the side walls, while the next flight was inverted and acted to recombine the streams towards the centerline of the channel. The gaps between the flights also allowed for easy and unobstructed viewing of sperm motility. Channel dimensions and herringbone designs were based on previously published herringbone microfluidic mixers [33,44]. The main channel dimensions were 1 mm (W) × 25 μm (H) × 15 mm (L) and herringbone structure heights were 25 μm (H). Control devices containing no micromixers were fabricated for comparison, with the same fluidic layer height of 25 μm. Two sample inlet configurations were evaluated in this study: the split inlet and the T-inlet. For the split inlet design, the sperm inlet was divided into two ~70 μm wide channels and the water inlets were divided into three ~290 μm wide channels. The T-inlet only contained a single ~143 μm wide sperm inlet channel and two ~430 μm wide water inlets. 

### 2.2. COMSOL Modeling of Prototyped Designs

Flow characterization and mixing performance of chip designs were simulated using COMSOL Multiphysics 6.0 (COMSOL Inc., Stockholm, Sweden). Channel velocity, pressure, diffusion, and particle flow were determined for different designs and flow rates appropriate to sperm analysis. All simulations were run under the same boundary conditions and mesh parameters and evaluated using the nodes of the same results. “Controlled Diffusion Micromixer” and “Residence Time” simulations were used to create the initial boundary conditions and simulation settings used (COMSOL Inc., Stockholm, Sweden). Geometry was imported from Inventor using LiveLink and water (liquid) was chosen as the material. 

The COMSOL “Transport of Diluted Species” interface was used with creeping flow in a stationary study. Pressure boundary conditions for the sperm and water inlet(s) were set to atmospheric with the outlet using flow rate. To correlate the simulation and experimental studies, fluorescein sodium salt solution (HiMedia Laboratories, Kennett Square, PA, USA) at 0.46 mM with a diffusion coefficient of 4.25 × 10^−6^ cm^2^/s was used [45,46]. Studies were performed at flow rates that were deemed appropriate for sperm activation in prior studies, 0.5, 1, and 5 μL/min. Cut line and cross-sectional planes at regular intervals spaced along the main channel were used to extract data such as concentration and velocity. The last cut line was used to calculate Reynolds (*Re*) and Pèclet (*Pe*) numbers according to the following formulas:(1)$$Re=\frac{\rho uL}{\mu },$$
where *ρ* is fluid density, *u* is the fluid velocity, *L* is the characteristic length (e.g., main channel width or 1 mm), and *µ* is the dynamic viscosity.
(2)$$Pe=\frac{Lv_{x}}{D},$$
where *L* is the characteristic length (e.g., main channel width or 1 mm), *v_x_* is the fluid velocity, and *D* is the diffusion rate. 

To quantify the mixing efficiency of the designs, the following formula was used from previous work [47]:(3)$$Mixing\;efficiency=\left(1-\frac{\sqrt{\frac{1}{N}\sum_{i=1}^{N}{\left(I_{i}-I_{i}^{Perf.mix}\right)}^{2}}}{\sqrt{\frac{1}{N}\sum_{i=1}^{N}{\left(I_{i}^{0}-I_{i}^{Perf.mix}\right)}^{2}}}\right),$$
where N is the total number of pixels, $I_{i}$ is the fluorescence intensity at pixel i, $I_{i}^{0}$ is the intensity at pixel i with no mixing or diffusion, and $I_{i}^{Perf.mix}$ is the intensity of the perfectly mixed solution at pixel i. The data for a perfectly mixed solution were obtained by placing or assigning the same diluted fluorescent solution in all inlets.

### 2.3. Fabrication and Preliminary Testing

#### 2.3.1. Soft Lithography Fabrication of Chips

Soft lithography was used to fabricate the microdevices, where AutoCAD (Version Q.111, Autodesk, San Rafael, CA, USA) was used to draw the fluidic and herringbone designs. Photomasks were printed on transparency films by a commercial provider (CAD/Art Services, Inc., Brandon, OR, USA). Using the photomasks, fluidic and herringbone master molds were created, with photoresist SU-8 2025 (Kayaku Advanced Materials, Westborough, MA, USA) by use of a single-layer photolithography process. A two-layer photolithography process was also evaluated for the study design. Briefly, SU-8 was spin-coated (Laurell Technologies Corporation, North Wales, PA, USA) on a silicon wafer (University Wafer, Inc., South Boston, MA, USA) with a thickness of 25 μm, UV cured (American Ultraviolet^®^, Lebanon, IN, USA) with a photomask and developed (to remove unexposed SU-8) with SU-8 Developer (Kayaku Advanced Materials, Westborough, MA, USA). The wafers were cleaned with isopropyl alcohol (IPA, ≥99%, VWR International, Radnor, PA, USA) and deionized water (DI water, ≥17.8 mΩ) and dried with nitrogen gas. For the two-layer approach, an additional 25-µm layer was spun superficial to the first layer, and the second photomask was aligned and exposed. The two-layer master mold was developed following the same steps as the single-layer mold.

A 10:1 mixture (elastomer:curing agent) of Sylgard 184 polydimethylsiloxane (PDMS, DOW Corning, Inc., Midland, MI, USA) was cast onto the molds, degassed in a vacuum chamber, and cured in an oven at 65 °C for at least 2 h. PDMS was de-molded from each wafer, cut to size using a razor blade, cleaned with IPA and DI water, followed by drying with nitrogen. 

To prepare the study design, fluidic channel and herringbone mixer geometry layers were aligned manually using an inspection scope (AmScope, Irvine, CA, USA). Ports were marked and holes were punched on the herringbone layer with 3 mm (inlet) and 1.5 mm (outlet) biopsy punches (INTEGRA Biosciences, Hudson, NH, USA). The layers were irreversibly bonded to each other after treating with oxygen plasma with a Harric Plasma Cleaner, PDC-32FG (Harric Plasma, Ithaca, NY, USA) for 45 s at 1.8 W and rested for at least 24 h prior to silanization. To silanize the channel and limit cell adhesion, a bonded chip was placed in the plasma cleaner and treated for 90 s. The chip was removed from the system and ~0.2 mL of 2-[methoxy(polyethlyeneoxy)6-9propyl]trimethoxysilane (Gelest, Inc., Morrisville, PA, USA) was injected into the outlet to fill the channel for 30 min, before flushing with DI water and drying with nitrogen gas. To prepare the fluidic-only control device that had no micromixer geometries, the same steps were performed with the fluidic microchannel layer bonded to a 25 × 75 × 1 mm glass microscope slide (Thermo Fisher Scientific, Hampton, NH, USA).

#### 2.3.2. Device Setup and Operation

The chip was pre-wetted by manually pushing DI water through the outlet using a 1-mL syringe. Excess water was removed from the inlets using a micropipette. A 1-mL plastic syringe (Henke-Ject, Germany) and ~5 cm of polytetrafluoroethylene tubing (PTFE, 0.0762 cm ID, 0.158 cm OD, MicroSolv Technology Corporation, Leland, NC, USA) were prefilled with DI water to prevent air bubbles in the channels. The chip was placed onto the microscope stage (Nikon Eclipse Ti2, Melville, NY, USA) and connected to the syringe pump by inserting the filled tubing into the outlet port. The syringe pump (UMP3-T1, World Precision Instruments, Sarasota, FL, USA) was placed ~5 cm away from the microscope stage vertically and horizontally to follow the natural curvature of the tubing and prevent any kinks or harsh angles within the connection. Sample inlets were loaded with 15 µL of DI water or 15 µL of fluorescein sodium salt solution (HiMedia Laboratories, Kennett Square, PA, USA) at 0.46 mM to visualize mixing. 

#### 2.3.3. Experimental Evaluation of Prototyped Designs 

The syringe pump was started in the withdrawal direction to pull fluid into the chip at the given flow rates and reached steady flow within 5 min. Fluorescent images were collected using 10× magnification, 10% fluorescent lamp excitation energy, and a camera (pco.edge 5.5 sCMOS, PCO-Tech, Kelheim, Germany) exposure time of 50 ms. Mixing efficiencies were calculated using the steps outlined in Section 2.2. The experimental data for a perfectly mixed solution were obtained by distributing a pre-mixed fluorescein solution in all inlets. Calculations were conducted before the start of the mixing geometries (x = 0 µm), halfway through the mixing geometries (x = 4850 µm), and after the last set of mixing geometries (x = 10,000 µm). Pixel intensity profiles across the channel were extracted from raw images taken using ImageJ (version 1.52, National Institutes of Health).

### 2.4. Application of the MAGS 

#### 2.4.1. Fish Husbandry

Zebrafish sperm were used to evaluate the functionality of prototypes. Protocols for the use of animals in this study were reviewed and approved by the Louisiana State University Institutional Animal Care and Use Committee. Adult zebrafish were maintained at the Aquatic Germplasm and Genetic Resources Center (AGGRC) in a recirculating system. Target water quality parameters were 20–26 °C, pH 7.5–8.5, and 14 h light:10 h dark photoperiod. Fish were fed twice daily with a dry food mix (zebrafish.org/documents/protocols/pdf/Fish_Feeding). Additional water quality parameters were monitored weekly and maintained within an acceptable range included: ammonia (0–1.0 mg/L), nitrites (0–0.8 mg/L), and nitrates (0–15 mg/L).

Sperm were collected by the use of a protocol previously described [48]. Briefly, male fish were anesthetized with 0.01% Tricaine methanesulfonate (MS-222, Western Chemical, Inc. Ferndale, WA, USA), placed ventral side up on a moist sponge, and stripped by gently pressing abdominal areas with fingers. Sperm was collected into a 10-µL glass capillary tube (Drummond Scientific, Broomall, PA, USA), and immediately released into a 1.5-mL microcentrifuge tube containing Hanks’ balanced salt solution (HBSS, 0.137 M NaCl, 5.4 mM KCl, 1.3 mM CaCl_2_, 1.0 mM MgSO_4_, 0.25 mM Na_2_HPO_4_, 0.44 mM KH_2_PO_4_, 4.2 mM NaHCO_3_, and 5.55 mM glucose, pH 7.2) at 300 mOsmol/kg. Sperm concentration was initially adjusted to 1.0 × 10^8^ cells/mL based on a Makler^®^ counting chamber (TS Scientific, Perkasie, PA, USA) as base suspension. 

#### 2.4.2. Recording Zebrafish Sperm Activation

Approximately 5 cm of PTFE tubing was connected to the outlet of the chip and secured with a ring of UV-curable resin (Bondic, Niagara Falls, NY, USA). The chip was pre-wet by manually pushing DI water through the outlet using a 1-mL syringe and the excess water was pipetted out of all inlets. The three water inlets were pre-wetted with 10 μL of DI water and the two sample inlets were loaded with 10 μL of 1.0 × 10^7^ cells/mL zebrafish sperm solution. A micropipette was set to 5 μL, suppressed to the first stop, and inserted into the tubing. The plunger was slowly released to pull flow for 3 s and removed from the tubing after 5 s. Zebrafish sperm were viewed with a dark-field microscope at a magnification of 200-x (CX41; Olympus, Tokyo, Japan), and images were captured with the charge-coupled device camera of the CASA system (HTM-CEROS, version 14 Build 013; Hamilton Thorne Biosciences, Beverly, MA, USA) using the Animal Motility routines provided [48]. 

### 2.5. Statistical Analysis

Statistical analyses were performed using GraphPad Prism (v8, GraphPad Software, San Diego, CA, USA). A single-tail *t*-test was performed to determine significance of recorded values indicated in the figure legend. *p*-values of <0.05 were considered significant.

## 3. Results

### 3.1. Design of Inlet and Passive Micromixer Geometries

A control device containing only a fluidic layer with a single inlet and no micromixing geometries was used as the baseline for all simulations and experimental studies. The split inlet design increased the surface area between the aqueous streams with a minimal pressure drop across the device. The proposed micromixer geometries, adapted from the standard herringbone [33], were designed specifically to be used with the split inlet design. The symmetric patterning used in the MAGS geometry was designed to ensure even sample distribution and homogenous mixing across the width of the channel.

### 3.2. COMSOL Modeling of Prototyped Designs

COMSOL simulations provided preliminary data for the design performance of micromixer and inlet geometries. To distinguish the sperm inlets from the fluidic inlets, sperm sample inlets were assigned an initial concentration of fluorescein used in the experimental studies to validate mixing. Diffusion of the sperm inlet stream into the fluidic stream was chosen to model the water transport and change in osmotic pressure that leads to motility activation in zebrafish sperm [49].

The first simulation involved the single inlet design, which focused on the fluorescein stream down the center of the main channel, flanked by the aqueous streams. When no micromixer geometries were present, mixing was limited, as evidenced by the distinct “No Mixer” concentration profiles (Figure 3). Reducing the flow rate from 5.0 µL/min to 0.5 µL/min allowed more time for diffusion into the adjacent streams. However, minimal diffusion occurred as evidenced by the continued persistence of the fluorescein stream. The addition of micromixers, represented by “Symmetric Mixer” and “Standard Mixer” in the figure, directed the fluorescein inlet stream to include a convective element to the mixing beyond diffusion alone. Due to the single inlet design and the design of the “Symmetric Mixer”, the fluorescein stream was unable to be split and recombined as intended. When the stream reached the second set of herringbones, there was a higher degree of mixing when compared to the “No Mixer” case. With a single inlet, the “Standard Mixer” design performed best, compared to the “No Mixer” and “Symmetric Mixer” designs at initiating and improving mixing. The “Standard Mixer” design allowed for disruption and directing of the fluorescein stream at the beginning of the channel and was more homogenously dispersed across the width of the channel, compared to the other two designs. 

Concentration values were calculated at different cut lines along the main channel of the single inlet designs (Figure 4). These results align with the visual observations made from concentration profiles (Figure 3). Diffusion occurred in the “No Mixer” case, with none of the flow conditions resulting in uniform distributions across the channel width. As the flow rate increased, the concentration curves narrowed, which was also observed in the concentration profiles (Figure 4). Introduction of the micromixer geometries resulted in increased fluorescein distribution at cut lines 2–5 when compared to the “No Mixer” case, and a homogenous concentration of ~0.06 mol/m^3^. Due to the single sperm inlet, the “Standard Mixer” produced more diffusion along the channel when compared to the “Symmetric Mixer” case, which was also observed in the concentration profiles. Cut line 1 was similar across all tested designs, with the peak concentration increasing directly with the flow rate. The Reynolds (Re) numbers calculated for all flow rates simulated were <0.003, and Pèclet (Pe) numbers >>1, indicating mass transfer was convection-dominated rather than diffusion-dominated in these scenarios.

The second set of simulations evaluated the split sperm inlet design, which utilized two sperm inlets and three water inlets to increase the surface area contact between the streams. This resulted in an increase in the distribution of fluorescein across the channel for all designs and studies (Figure 5), compared to the single inlet design. The largest increase was observed when comparing the “No Mixer” designs of the single inlet and the split inlet configurations. Considering the “Symmetric Mixer” and “Standard Mixer” cases, it was evident that splitting the sperm inlet streams to align with the vertices of the “Symmetric Mixer” herringbone peaks resulted in the streams being split and recombined as intended. The “Symmetric Mixer” design concentration profile was more symmetrical across the channel width when compared to the “Standard Mixer” design.

Concentration plots (Figure 6) along the main channel validated the visual observations (Figure 5). As seen in the single inlet design, both micromixer geometries improved mixing within short distances down the main channel. The “Standard Mixer” design, however, resulted in a non-uniform distribution of channel concentrations. The “Symmetric Mixer” design showed similar mixing performance, compared to the “Standard Mixer” design, but maintained a homogenous channel concentration profile. All designs showed similar concentration profiles at cut-line #1, the entrance area prior to mixers, but the “Symmetric Mixer” and “Standard Mixer” designs had more distributed concentration profiles for cut lines #2–5 when compared to the “No Mixer” design. The calculated Re and Pe numbers were slightly increased, but comparable to the single inlet design (Figure 3), they showed no change with the introduction of passive mixers and were positively correlated to the flow rates. Re numbers were all <0.002 and Pe numbers were >>1.

Because evaluating sperm motility in microfluidic channels can be more challenging when the velocity of the fluid in the viewing area of the channel is non-uniform, velocity profiles were compared for the inlet and mixer configurations. The devices with the split inlet channel configurations had similar velocity profiles in all evaluated flow rates (Figure 7). However, the “Symmetric Mixer” exhibited higher velocities near the walls for cut lines 2–4, while the “No Mixer” case did not. Both “Standard Mixer” designs, single or split inlets, had greater fluctuations in the velocity profiles (±1000 μm/s) for all simulations. However, the velocity ranges for all designs were similar (~4000–8000 μm/s for 5.0 μL/min, ~1000 μm/s for 1.0 μL/min, and ~500 μm/s for 0.5 μL/min). 

Mixing efficiencies were calculated using the COMSOL diffusion data for the three flow rates evaluated (Figure 8). As expected, mixing efficiency increased along the main channel, and the addition of micromixer geometries further increased mixing efficiency. This was more evident with the single inlet (~30–70%, depending on flow rate) when compared to the split inlet design (~60–80%, depending on flow rate). Splitting the sperm inlet increased efficiencies for all evaluated designs and flow rates (5–31% increase) except for the “Symmetric Mixer” design (~9% reduction). Overall, mixing efficiencies for the split inlet design was similar for the “Symmetric Mixer” and “Standard Mixer” at the evaluated flow rates.

### 3.3. Fabrication and Preliminary Testing

#### 3.3.1. Device Setup and Operation

The device design allowed for operation on either an inverted or standard microscope with a syringe pump (steady state flow) or with a laboratory micropipette (step flow). Throughout the experiments, the pump and pipette were each able to operate the device under negative pressure, which allowed the sperm sample to be easily pipetted directly into the inlet ports and drawn into the device. With the syringe pump, various flow rates were evaluated, which facilitated mixing evaluation via fluorescence microscopy. To demonstrate the ease of use in a typical biology laboratory setting, a standard micropipette was also used to draw fluid through the device. This mode of operation resulted in non-steady-state flow, where a pulse of sperm flowed through the channel with velocity magnitude and duration dependent on the set-pull volume and plunger-actuation velocity of the micropipette. Flow cessation was achieved for both the pump and pipette mode of operation, in less than 1 s, by disconnecting the tubing from the outlet.

#### 3.3.2. Experimental Evaluation of Prototyped Designs 

Fluorescence microscopy images of fluorescein distribution in the devices were comparable to the COMSOL simulation results. An increase in flow resulted in a narrowing of fluorescent streams, leading to less lateral dispersion across the fluidic channel. At the lower flow rate of 1 μL/min, the fluorescein completely dispersed across the channel width by the end of the channel length (Figure 9). An increase to 5 μL/min flow rate narrowed the fluorescein streams at all three regions and resulted in minimal diffusion. The line plots extracted from the images agreed with the visual observations of fluorescein distribution. The inclusion of micromixer geometries resulted in increased dispersion at earlier points along the main channel. As with the fluidic device, increasing the flow rate narrowed the fluorescein streams at the beginning, but the micromixer structures forced convective dispersion of the dye distribution across the channel. The resulting line plots from the MAGS device incorporating the “Symmetric Mixer” design showed a lower intensity at the middle and end of the channel when compared to the “No Mixer” control device. 

Mixing efficiencies of the experimental data (Figure 9) and the COMSOL simulation data (Figure 4 and Figure 6) showed changes when mixing geometries were added. The “Symmetric Mixer” increased mixing efficiencies for the COMSOL and experimental data by at least 15%, with increases as high as 35% based on the flow rate and location in the channel. The experimental data showed a larger increase in mixing efficiencies with the addition of the micromixers. Experimental mixing efficiencies agreed with the COMSOL results for all scenarios but were lower depending on the flow rate and mixing features, likely due to the limited sensitivity of the fluorescence microscopy system in detecting minute variations of the fluorescein probe, assumption, and boundary conditions placed in COMSOL simulations and variations in the device setup and operation.

### 3.4. Zebrafish Sperm Activation Using the MAGS

A sample of zebrafish sperm was split to be compared in the MAGS device as well as a Makler® chamber, a static counting device that is a current standard for CASA, to observe zebrafish activation. As sperm were drawn into the main channel of the MAGS, where they could be viewed via the microscope, flow cessation occurred within 1 s after disconnecting the pipette from the device. Sperm activation was achieved in the device, where motile sperm could be observed at locations. The MAGS device was compatible with the CASA system and showed comparable sperm motility to the manual activation observed with the Makler® chamber (Figure 10). 

## 4. Discussion

The reliable handling of sperm from imperiled species or valuable biomedical models (e.g., zebrafish) hinges on consistent motility activation techniques that can be used to evaluate sperm quality. Commercially available devices, such as the Makler^®^ counting chamber, are costly and depend on manual mixing with a micropipettor by a technician which can lead to variability between users. Technological developments such as CASA have reduced human error by implementing automated tracking software [36,49]. However, the activation of sperm is still subject to variation in user technique, hindering standardization and reproducibility. Microfluidics technology offers rapid and efficient on-chip mixing of samples and has been widely used in other scientific applications but has not yet been widely applied for sperm activation and analysis. Thus, this work evaluated various microfluidic designs for the development of a high-dilution, novel micromixer geometry to standardize the activation and motility assessment of zebrafish sperm. 

The channel width dimensions of the MAGS device were maximized to facilitate sperm sample entry into the micromixer geometries and also permit the main channel to serve as a viewing chamber. The channel heights were restricted to 25 µm to prevent excess stacking of the sperm that could complicate motility analysis in microscopy systems with a large depth of field. The configuration of the sperm sample inlets and the diluent streams were designed to maximize the dilution ratio. Previous sperm microfluidic devices used a 1:1 or 1:2 ratio of sperm to diluent with either a two-inlet or three-inlet configuration [36,38]. Because the motility of zebrafish has been shown to vary within the first minute of activation [48,49], a single-step dilution was preferred over a serial-staged dilution array. By splitting the sperm sample into two inlets, each flanked by a center and two outer water inlets, there was a greater interfacial area between the sample and the diluting (and activating) water streams. The dimensions of the sperm and water inlets were also optimized to reach a final dilution of 1:6, which can yield greater sperm activation than lesser dilution ratios [50]. 

The design of the micromixer geometry was the second area of focus of this study and was important for rapid sperm activation and positioning within the microchannel for motility analysis. The passive mixing geometry was adapted from previously published designs but was modified to best interact with the configuration of the inlet streams [33,34,44]. For optimal mixing, it was important for the sperm streams to align with the upstream vertices of the v-shaped herringbone structures. At the intersection of the inlet streams and main channel, the micromixing chevron structures acted to initially split and then recombine the sample streams, forcing fluid mixing and generating higher mixing efficiency than the standard herringbone geometries. The “Symmetric Mixer” chevron designs have two v-structures that initially split both sample inlet streams towards the outside of the channel, and on subsequent flights are reversed in orientation to recombine the sample streams. To facilitate observation of sperm motility, these latter reversed-chevron designs ensured that sperm were focused toward the center of the channel rather than at the periphery where wall interactions could alter normal swimming behavior, as documented previously [51]. 

The mixing efficiencies of the MAGS device were comparable to other previously published micromixers. Park et al. evaluated downstream distance from the intersection to achieve mixing in their herringbone design at flow rates 0–4 µL/min, where 5.6–8.4 mm was needed to achieve >90% mixing efficiency [36]. Scherr et al. simulated mixing efficiencies for three common geometries compared to their novel micromixer, where at the lowest Re evaluated (~1), their SeLMA design showed the highest performance (~95%) [37]. Beckham et al. compared two different versions of this SeLMA micromixer with the mixing efficiency increasing from 57 ± 6% to 94 ± 1% as the flow rate decreased from ~9 to ~3 µL/min [38]. Even though the MAGS device was operated at much lower Re numbers (<<1) compared to these studies >50% efficiency was achieved very quickly after the first flight of symmetric mixers (1.8 mm from inlet). Halfway down the channel, 4.85 mm from the inlet, the mixing efficiencies were 76% (1 µL/min) and 68% (5 µL/min). These remained relatively unchanged towards the end of the device, 79% (1 µL/min) and 72% (5 µL/min).

There was ample length between each flight of chevron structures to monitor motility in a full field of view compatible with either a standard 100-x microscope, evaluated in inverted and non-inverted configurations or the Hamilton-Thorne HTM-CEROS CASA system used in many motility studies [3,48,49]. The use of multiple viewing areas between the chevron flights down the channel could enable the study of a single sample along a gradient of time within the same MAGS device, which could be cumbersome in standard microscope slide chambers. Approximately halfway down the main channel or after the third flight of micromixer structures, the highest-level motility was observed. Future iterations of the device could further reduce the number of mixing flights and channel length, reducing the overall volume required to flow in the device. This would make the MAGS more useful for imperiled or small-bodied species where sample volumes are minuscule and can be difficult to acquire [52].

Computational fluid dynamic tools are an excellent and common method used to simulate the function of micromixer geometries and evaluate performance [53,54]. A design change for the MAGS device, compared to previous microfluidic devices, was the option to operate the chip with negative pressure (pulling fluid through the device) instead of positive pressure (pushing fluid through the device). Simulation within the device showed lesser pressure drops than reported previously for microdevices that contained constrictions within the channel, making initial sample injection difficult and contributing to longer flow cessation times [36,38]. The benefit of this versatility is that end-users without access to syringe pumps may be more inclined to use this device and its simpler mode of operation. Research applications could still utilize steady-state operation with positive pressure that may allow for greater applicability outside of initial motility assessment. For example, toxicity studies that would require a steady flow of sperm to be rapidly mixed with fast-acting compounds could be performed in the short term in the upstream viewing areas or downstream ones for extended exposure studies [55].

Fabrication of the MAGS device involved traditional photolithography techniques and materials that are well documented in sperm handling for aquatic repository applications with no adverse side effects [38,56]. The complexity of the micromixer design necessitated two-layer lithography, which is challenging due to the precise alignment required for the micro-scale features. While plasma bonding between a PDMS chip and a glass slide is more conventional, PDMS–PDMS bonding is a relatively simple and effective alternative that circumvents the tough requirements of two-layer lithography [57]. This approach allowed the molds for the inlets and main channel geometries and herringbone mixing structures to be rapidly prototyped as single-layer molds rather than creating the more complex two-layer photolithography mold for each revision. For example, the inlet configuration used a separate reservoir for each inlet whereas other devices have shared reservoirs to make plumbing and flowing more consistent when using positive pressure. That approach required the alignment of the two PDMS layer casts for device assembly, but this was readily accomplished in this study without any alignment equipment. 

A key advantage of the MAGS device is that it permits the measurement of subsequent trials of motility from a singular sample, whereas a static chamber can only measure one instance of motility. In the static chamber, the entire sample is activated, whereas, in the MAGS device, sperm in the inlet channels do not activate until exposed to water in the micromixer region of the device. More extensive evaluation of motility analysis of zebrafish and other aquatic species sperm in the MAGS device is planned in future studies. The MAGS device could facilitate these studies, and in particular, reinforce the recommendations on best approaches for zebrafish motility analysis recently made by Blackburn et al. [48]. It is hoped that the MAGS device could enable earlier measurements of motility so that a more complete time course over standardized timepoints could be accomplished, rather than simply reporting peak motility at inconsistent times, which makes comparison of data across laboratories difficult.

## 5. Conclusions

The success of aquatic species research laboratories and germplasm repositories requires accurate and reproducible methods to activate and evaluate sperm samples. Motility assessment is a crucial and easy evaluator of sperm quality which directly influences studies such as cryopreservation and fertilization. The MAGS device, with an inlet channel configuration designed for a high dilution ratio and a novel micromixer design, served to minimize the difficulty and human error often associated with manual sperm activation and motility assessment. Together, the inlet configuration and novel micromixers resulted in increased mixing efficiencies, homogenous flow, and proper cell positioning in the channel and could be viewed with a standard microscope or commercially available CASA systems. The MAGS device can be operated in either positive or negative-pressure modes and showed sperm activation comparable to that achieved with hand mixing and viewing in a standard Makler^®^ counting chamber.

Future iterations of the MAGS device could be modified to meet the requirements of other species with sperm requiring activation via dilution, for instance in marine species where various dilutions of seawater are used or in species with larger sperm sizes. While soft lithography techniques have been the most viable fabrication method of devices at this scale, it requires skilled labor, uses cost-prohibitive equipment, and takes considerable time [56,58]. Increases in 3D printing resolution are now making this an attractive alternative for the fabrication of microfluidics devices at a fraction of the cost (<USD400), time, and skill requirements [59]. The MAGS design could be modified and 3D printed relatively easily for less expensive and more rapid device fabrication.

## Figures and Tables

**Figure 1 micromachines-14-01310-f001:**
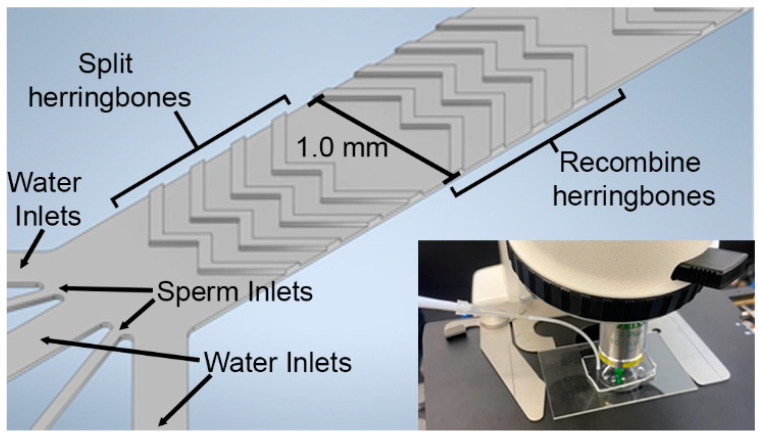
Detailed design of the Micromixer for Assessing zebrafish Sperm quality (MAGS) inlets, main channel, and symmetric herringbone geometries. The main channel width is 1.0 mm across, channel height and herringbone height are each 25 μm. The bottom right corner image depicts the MAGS setup and operation using a micropipette to pull sample into the device, where sperm motility was evaluated using video microscopy.

**Figure 2 micromachines-14-01310-f002:**
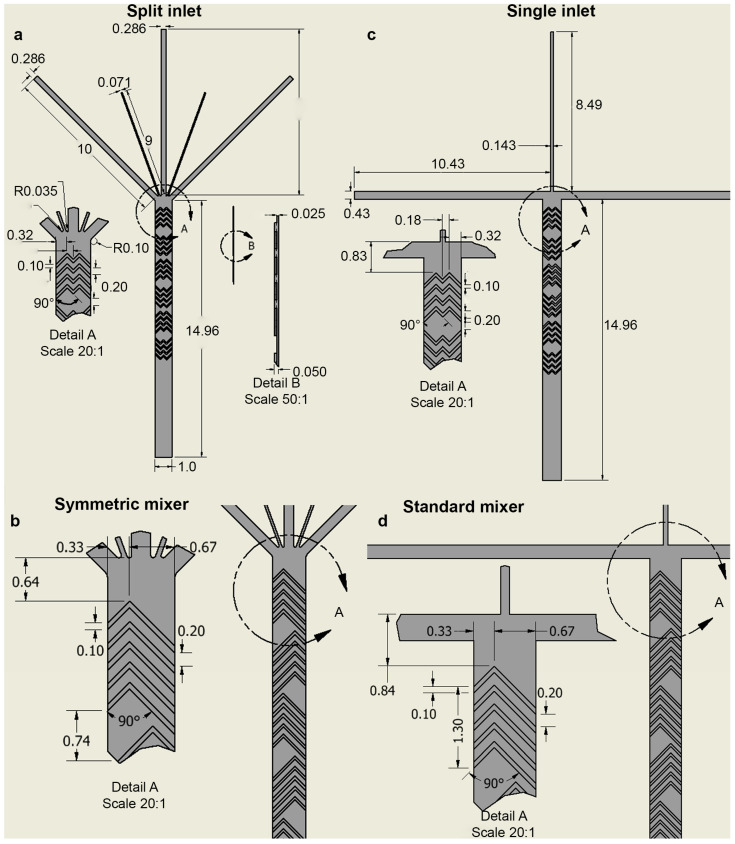
Design schematics of the “Symmetric mixer” geometry and inlet array used to create the tested microfluidic devices. The MAGS device included (**a**) the novel split inlet design and “Symmetric mixer” geometries and was compared to (**b**) a “Standard mixer” geometry for mixing efficiency. Single inlet versions for (**c**) the “Symmetric” and (**d**) “Standard mixer” geometries were compared to the split inlet design. All dimensions shown in millimeters.

**Figure 3 micromachines-14-01310-f003:**
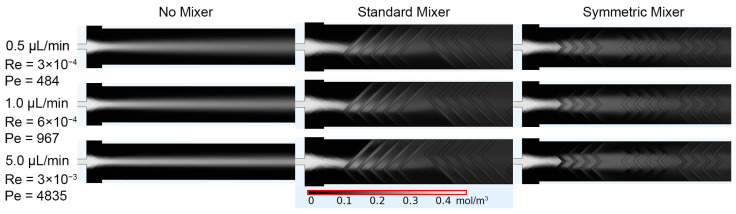
COMSOL generated fluorescein concentration profiles for the single inlet designs of No Mixer, Standard Mixer, and Symmetric Mixer simulations, at 0.5, 1.0, and 5.0 μL/min. Average Re and Pe values were calculated after the last set of mixers in the main channel (x = 10,000 μm) and reported under the corresponding flow rates.

**Figure 4 micromachines-14-01310-f004:**
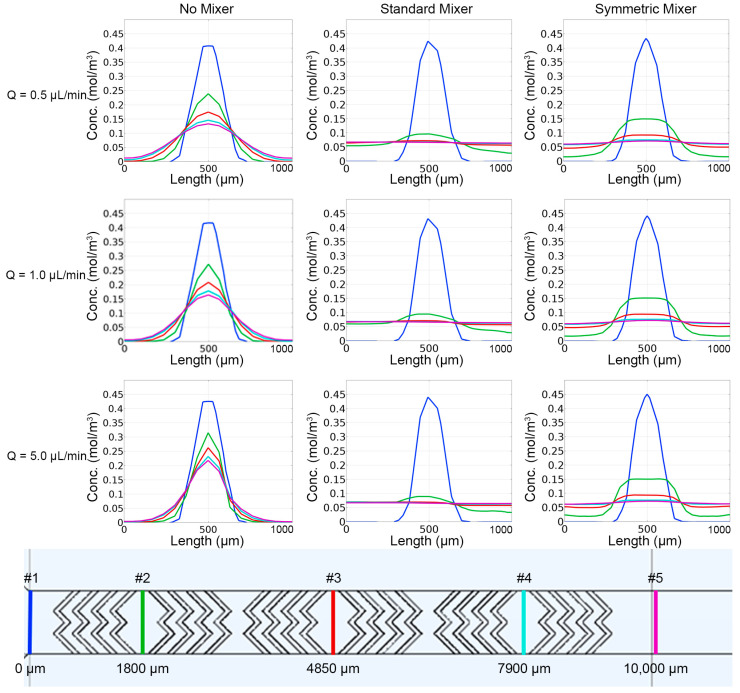
Concentration profiles produced from COMSOL simulations for the single inlet designs (**top**). Location of cross-sectional cut-lines used to as evaluation sites, with color of the line matching the concentration profiles above (**bottom**).

**Figure 5 micromachines-14-01310-f005:**
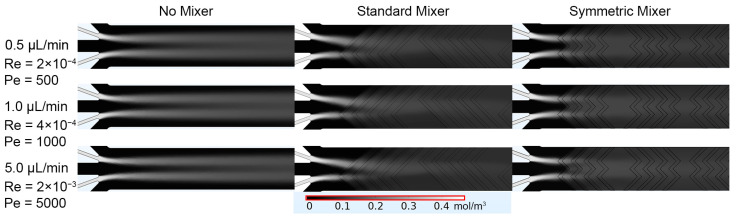
COMSOL generated fluorescein concentration profiles for the split inlet designs of No Mixer, Standard Mixer, and Symmetric Mixer simulations, at 0.5, 1.0, and 5.0 μL/min. Average Re and Pe values were calculated after the last set of mixers in the main channel (x = 10,000 μm) and reported at the corresponding flow rates.

**Figure 6 micromachines-14-01310-f006:**
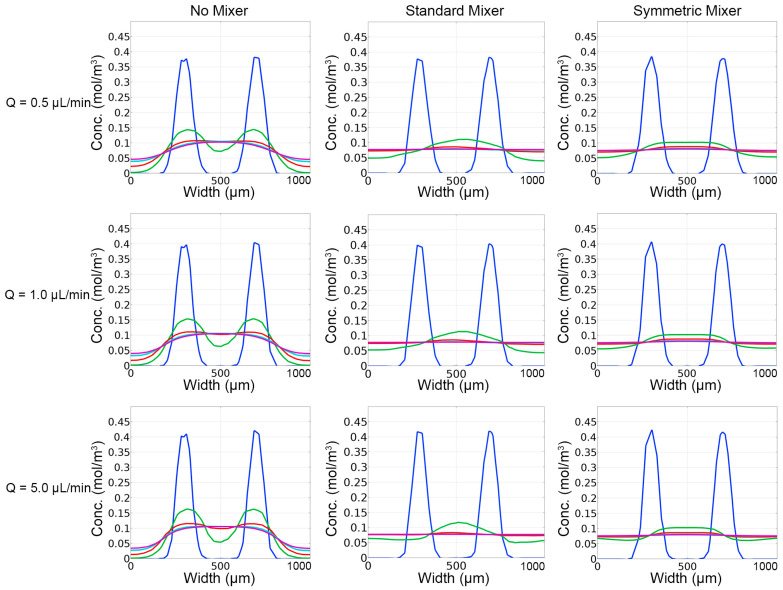
Concentration profiles produced from COMSOL simulations for the split inlet designs. The location of cross-sectional cut-lines used to as evaluation sites, with color of the line matching the concentration profiles above are shown in Figure 4, bottom.

**Figure 7 micromachines-14-01310-f007:**
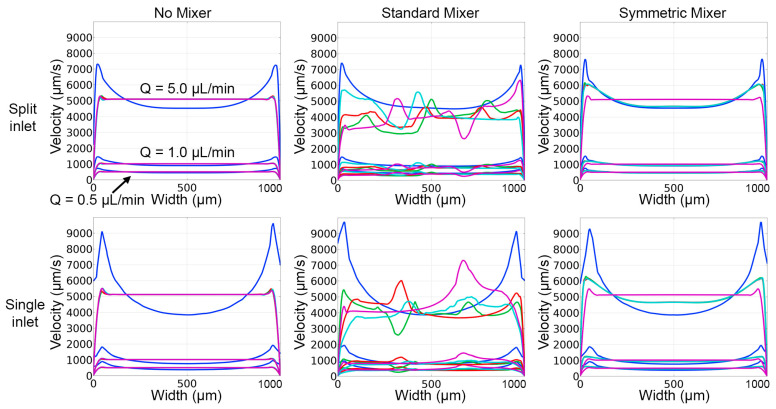
Velocity line plots produced from COMSOL simulations and concentration profile results for all evaluated designs. The location of cross-sectional cut lines used to as evaluation sites, with color of the line matching the concentration profiles above are shown in Figure 4, bottom.

**Figure 8 micromachines-14-01310-f008:**
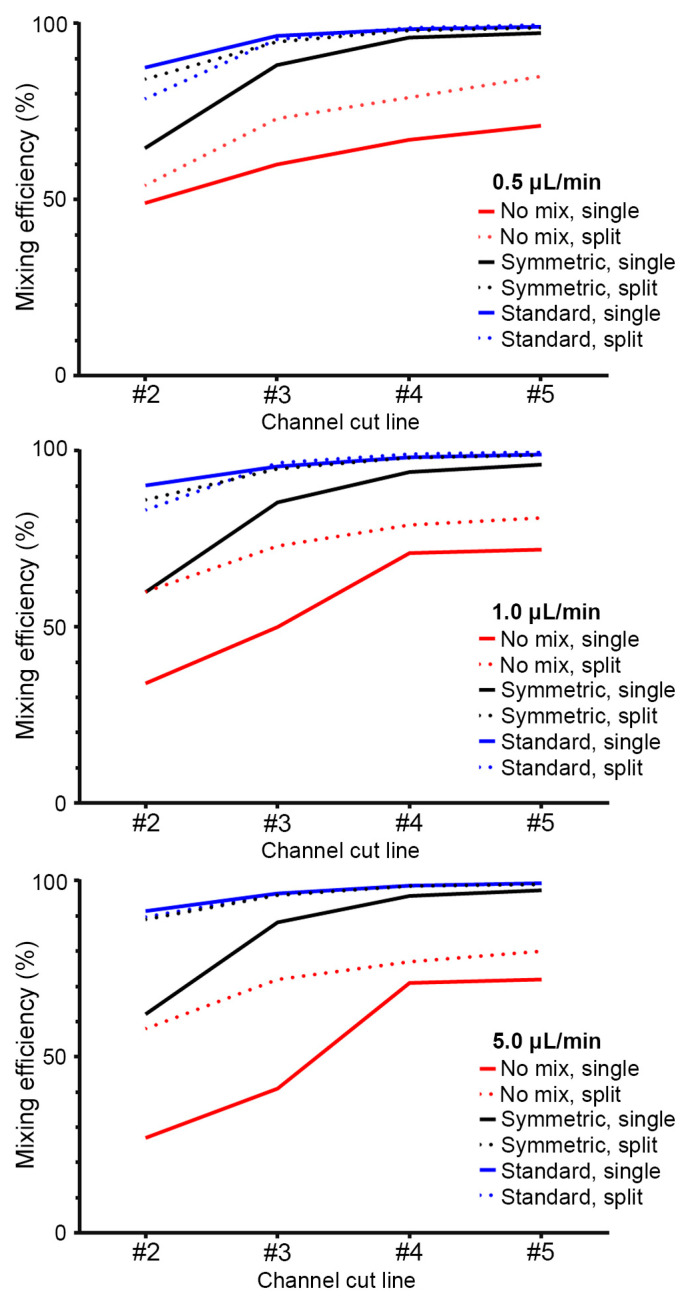
Mixing efficiencies from COMSOL concentration plots at each of the cut-lines down the microchannel, where cut-line #1 was taken as the basis for the mixing efficiency calculations. Single inlets are shown with solid lines and split inlets with dashed lines.

**Figure 9 micromachines-14-01310-f009:**
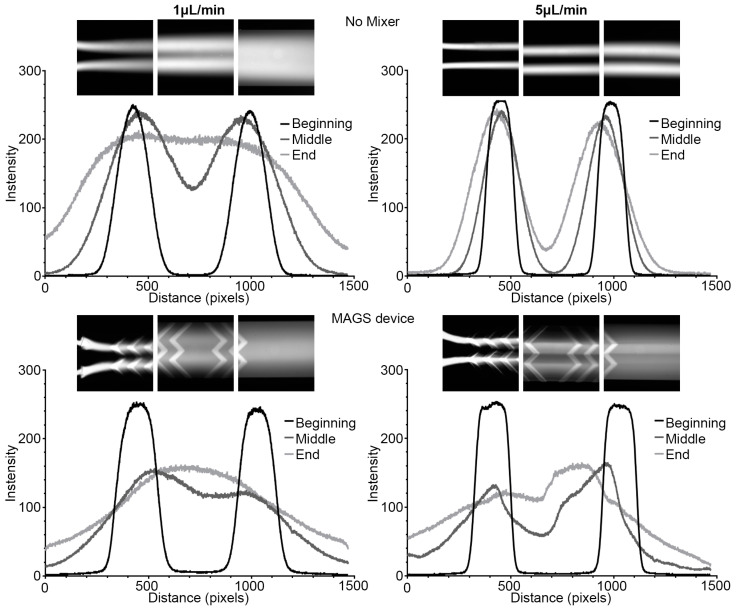
Experimental intensity plots for “No Mixer” and MAGS devices at 1.0 and 5.0 μL/min. The images used to obtain line plots are above each plot and data were extracted using ImageJ. Images were captured at the beginning (Cut-line #1), middle (Cut-line #3), and end (Cut-line #5) of the main channel.

**Figure 10 micromachines-14-01310-f010:**
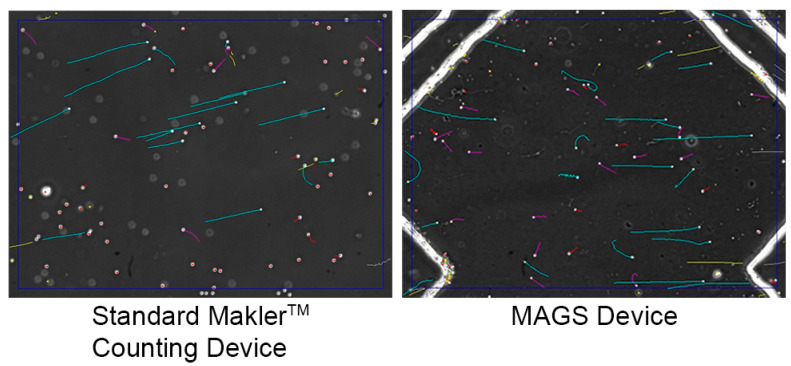
CASA motility tracking results of zebrafish sperm loaded into a Makler^®^ static counting chamber (**left**) and the MAGS device (**right**). Images contain sperm tracks from activity.

## Data Availability

All data is contained within the article.

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
