# Peer review of "A Modified-Herringbone Micromixer for Assessing Zebrafish Sperm (MAGS)"

_micromachines, 2023, doi:10.3390/mi14071310_

Round 1

Reviewer 1 Report

The manuscript reports a novel micromixer geometry that aligns inlet streams with modified herringbone structures to split and recombine flows. The work involves intensive simulation to study the concentration profiles, velocity, and mixing efficiencies, together with experimental validation to prove the feasibility. The work is solid enough to publish in Micromachines.

-        Still, I would suggest the authors improve the readability by shortening the manuscript and only presenting the main descriptions, discussions and conclusions in a journal paper style while the current version is very long, more like a report or thesis sections.

-        As the authors mentioned, the dilution ratio is a critical factor evaluating motility and this work has reached a very large dilution ratio of 1:6. I would suggest the authors provide this number in the abstract to clearly show the progress.

-        It would be great if the authors can create a list comparing the performance of this new device with previously published microdevices.

Reviewer 2 Report

This manuscript presented a modified herringbone micromixer to perform sperm motility analyses. The novelty of the manuscript lies in, through the design of the micromixer, i.e., using splitting inlet and symmetric herringbone microstructures, they achieve a dilution ratio of 1:6, which yields greater sperm activation. Overall, the manuscript is acceptable for publication. However, in my opinion, the logic of writing the manuscript or conducting the research is doubtful. This is because it seems the authors didn’t focus on their main novelty, i.e., how the optimized micromixer is helpful for the sperm activation. Instead, they took great effort to analyze the mixing efficiencies in numerical simulations and experiments. However, they didn’t present any result on the relations between mixing efficiencies and activation quantitatively. Lack of information in this aspect might lead to misunderstanding for readers, and the evidence and conclusion of this manuscript seem not quite relevant, although the authors presented a quite long discussion. I hope authors will improve their manuscript in this aspect in revision. My other concerns are:

1.     It would be nice if the authors report some quantitative comparison between numerical simulations and experiments.

2.     In simulations, the authors used three types of microchannels, no mixer, Study mixer (by the way, I don’t like this name, perhaps “symmetric herringbone micromixer” is better), and Standard mixer. Why only two of them are considered in experiments?

3.     The analyses from lines 386-404 is somehow boring. Actually, the readers might care more about the value of the mixing efficiencies, instead of the increasing rate. So, if I am right, the result is quite simple, i.e, symmetric herringbone micromixer with splitting inlet produces similar mixing efficiencies as the standard micromixer.

4.     Is it possible to measure the sperm mobility quantitatively?

5.     Some minor suggestions:

Line 152: “4.25e-6 cm2/s” is not the standard expression in scientific paper.

Line 159 and 162: “u is the fluid velocity,” “vx is the velocity,” “L is the characteristic length”. L and velocity should be explicitly defined.

The bottom figures in Figure 4 and 6 are repeated, not necessary.

English is qualified. 

Round 2

Reviewer 2 Report

The manuscript has been revised properly, and I think it is ready for publication.